# Come together: A unified description of the escalator capacity

**Christoph Gnendiger** [1]* , **Mohcine Chraibi** [1] , **Antoine Tordeux** [2]

**1** Institute for Advanced Simulation, Forschungszentrum Jülich, Jülich, Germany, **2** School of Mechanical Engineering and Safety Engineering, University of Wuppertal, Wuppertal, Germany

These authors contributed equally to this work.
* c.gnendiger@fz-juelich.de

**Data Availability Statement:** The open-source simulation software JuPedSim can be obtained from the Zenodo database (https://zenodo.org/record/6144559).

**Funding:** This work was supported by the German Federal Ministry of Education and Research

## Abstract

We investigate a variety of aspects related to the simulation of passenger dynamics on escalators, mainly focusing on the discrepancy between the 'theoretical' and the 'practical' capacity that is observed for these facilities. The structure of the paper is twofold. In the first part, we introduce a space-continuous model to describe the transition of agents from walking on the plain to standing on the escalator. In the second part, we use numerical findings from simulations to study important measures like minimum distances between the standing agents and average occupancies of the escalator steps. One of the most important results obtained in this paper is a generalized analytical formula that describes the escalator capacity. We show that, apart from the conveyor speed, the capacity essentially depends on the time gap between entering passengers which we interpret as human reaction time. Comparing simulation results with corresponding empirical data from field studies and experiments, we deduce a minimum human reaction time in the range of 0.15s–0.30s which is in perfect agreement with results from social psychology. With these findings, it is now possible to determine accurately the relationship between the capacity and the speed of an escalator, allowing a science-based performance evaluation of buildings with escalators.

## Introduction

With an ever-growing urbanization [1], the peak performance of pedestrian facilities in public buildings like shopping malls, entertainment centers, and sport stadiums, is gaining importance. Moreover, public transport via trains has become one of the most effective and climate-friendly transportation modes available. As a result, more multiple-level train and metro stations are being built and used on a regular basis. To increase comfort of the passengers as well as to satisfy in particular intermittent performance demands of buildings as a whole, escalators are being used ubiquitously as they generally transport large passenger groups much faster between different levels than stairs and elevators [2].

One important measure to assess and evaluate the performance of escalators is the maximum passenger flow that can be realized over a certain period of time. As with other pedestrian facilities, this handling capacity strongly depends on the geometrical dimensions which are

(BMBF) within the framework of the project KapaKrit under the grant number 13N14620. MC and AT acknowledge the Franco-German research project MADRAS funded in France by the Agence Nationale de la Recherche (ANR, French National Research Agency) under the grant number ANR-20-CE92-0033, and in Germany by the Deutsche Forschungsgemeinschaft (DFG, German Research Foundation) under the grant number 446168800. The funders had no role in study design, data collection and analysis, decision to publish, or preparation of the manuscript.

**Competing interests:** The authors have declared that no competing interests exist.

usually fixed and typically known or easily accessible. More importantly (and in contrast to many other pedestrian facilities), however, escalators have an additional setting option that can be adjusted dynamically during operation: the conveyor speed. According to this, the escalator capacity is not a fixed number but depends on the mode in which the facility is operated [3–5].

The question that naturally arises is how the escalator speed should be selected. There are two main points to consider. First, it is obvious that the choice of a particular conveyor speed is always a trade-off between comfort, safety, and performance demands as not all these aspects can be maximized at the same time. For instance, while a larger conveyor speed may increase the capacity, it also complicates the adaptation process of the entering persons to match the fast-moving escalator which in turn may endanger the passenger safety. The importance of the safety aspect is underpinned by the fact that escalator-related injuries have increased in the past years [6].

Second, in order to strike a balance between the aforementioned aspects, the performance of an escalator has to be known exactly. Surprisingly, however, there exists a long-standing and persistent ambiguity regarding the relation between conveyor speed and capacity of escalators. Many guidelines, norms, and handbooks, for instance, propose a *linear* speed-dependence of the maximum possible passenger flow [2,7–10]. The corresponding quantity is often referred to as 'theoretical' capacity although the underlying theoretical assumptions are often not explained in detail. At the same time, however, it is known that the idealized situation of maximum escalator utilization is almost never observed in field studies and experiments, see for instance Table 1 which summarizes empirical findings on maximum observed flows at escalators.

Regarding the discrepancy between 'theory' and observation, Fruin already noticed in the 70s the so-called 'empty'- or 'vacant'-step phenomenon which states that a fully utilized escalator is never observed, even in peak travel times [20]. Therefore, usually a so-called 'practical' or 'nominal' capacity is introduced in (manufacturer) handbooks and in norms. This quantity is consistently and significantly smaller than the 'theoretical' one. What often remains unclear, however, is the actual size and speed-dependence of the deviation as well as the underlying reason for it. Even among escalator manufacturers [3–5], there is no consensus about these points and, accordingly, many values for the 'practical' capacity can be found in the literature, see Fig 1 for a qualitative illustration. As a consequence, manufacturers and norms recommend speed-settings that might not be optimal.

In the following, we aim to contribute resolving this issue by putting the speed-capacity relation and therefore the performance analysis of escalators on a firmer basis. We do this by

**Table 1. References for maximum observed upward flows at escalators.**

| reference | escalator speed [ms$^{-1}$] | count interval [s] | maximum flow [s$^{-1}$] |
|---|---|---|---|
| Bodendorf et al. (2014) [11] | 0.50 | 10 | 1.73 |
| Böhm-Franke (2015) [12] | 0.50 | 10 | 1.80 |
| Kinsey (2011) [13] | 0.50 | 60 | 1.70 |
| Nai et al. (2012) [14] | 0.61 | 20 | 2.25 |
| Kahali et al. (2021) [15] | 0.65 | 30 | 2.57 |
| Kahali et al. (2021) [16] | 0.65 | 24 | 2.47 |
| Davis et al. (2002) [17] | 0.72 | variable | 1.98 |
| Al-Sharif et al. (1996) [18] | 0.75 | 30 | 2.03 |
| Majo (1966) [19] | 0.75 | 60 | 2.25 |

The clear width of the escalator is one meter in all cases. The column 'count interval' indicates the length of the time span in which the maximum flow is measured.

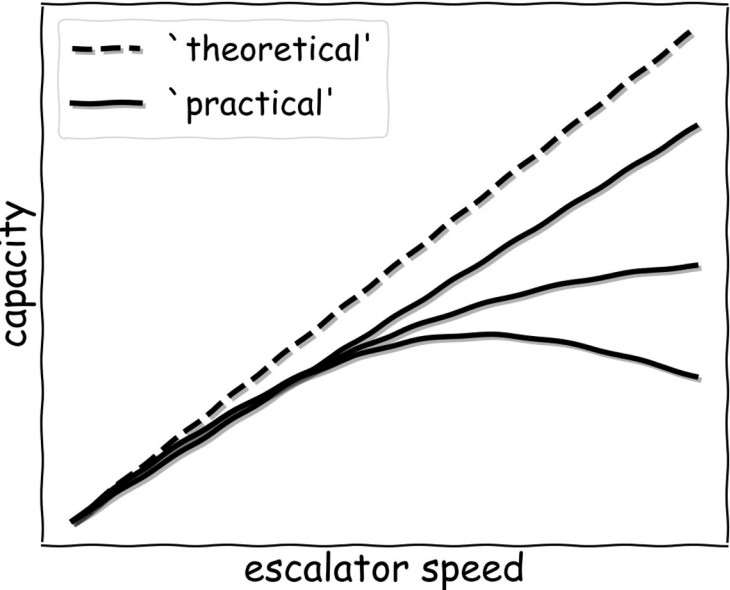

**Fig 1. Qualitative difference between the 'theoretical' and the 'practical' escalator capacity in the range of typical conveyor speeds.** Regarding the 'practical' capacity, linear, saturating, and even decreasing curve progressions can be found in the literature.

presenting an efficient space-continuous first-order model that describes the transition of agents from walking on the plain to standing on the escalator.

We suggest solving the duality problem of the capacity by proposing a generalized formula that

- is able to explain the deviation between 'theoretical' and 'practical' capacities,

- is at the same time able to explain the existence of different 'practical' capacities that can be found in the literature, and

- is compatible with observed empirical values from field studies and experiments.

Based on purely numerical studies, we deduce and validate an analytical formula that quantifies the variation of the escalator capacity with respect to its speed. Our results show that assuming standing agents leads to reasonable predictions of the escalator performance. Moreover, it is found that the obtained capacity formula is a valid generalization of the linear approach often assumed and used in the literature.

## Simulation of escalators

Usually, experimental and numerical studies of pedestrian dynamics serve as validation ground for mathematical models that aim to describe them. Concerning escalators, however, this approach is only partially practical as there is only little empirical data to compare with and due to limited possibilities to perform experimental work under extreme conditions and over an extended period of time. In this work, we therefore follow a different approach by developing a space-continuous model that describes the entering behavior of pedestrians at escalators. We implement the model in the open-source software *JuPedSim* [21] and capitalize on the obtained numerical results to formulate a theoretical concept of the escalator capacity. For the simulations, we use the geometry shown in Fig 2. In this setup, two floors with a height

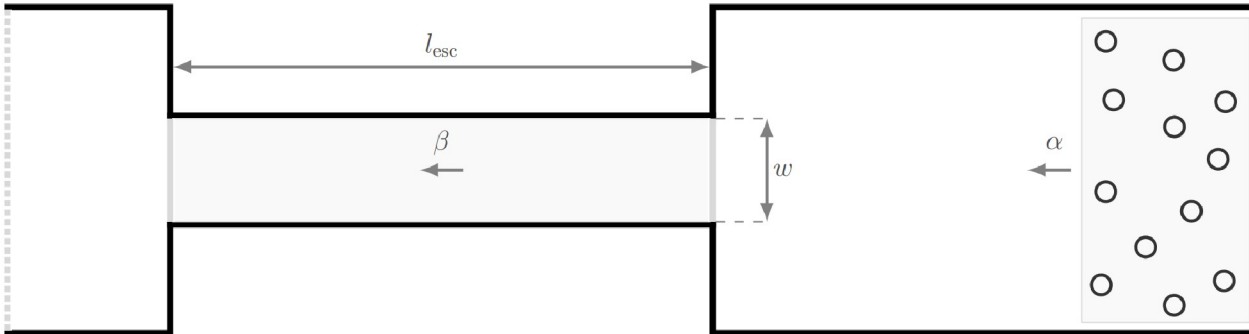

**Fig 2. Geometry used for the simulations.** The geometry consists of three parts: a lower floor including the agent sources (right), an escalator with a clear width of $w \in \{0.6, 1.0\}$m, and a projected horizontal length of $l_{esc} = 10$ m (middle), as well as an upper floor which is located $\Delta h = 5$ m above the lower one (left). The movement direction of the agents and of the escalator is from right to left. The incoming agent flow $\alpha$ can be freely adjusted whereas the flow $\beta$ is typically measured.

difference of $\Delta h = 5$ m are connected by a single escalator with a projected horizontal length of $l_{esc} = 10$ m and an inclination angle of approximately 27˚. As clear widths of the escalator, we consider the cases $w = 0.6$ m and $w = 1.0$ m. For the upcoming capacity analysis, it is important to generate a constant density in front of the entrance to the escalator. Accordingly, we ensure that the relation $\alpha > \beta$ holds in all simulations, where $\alpha$ is the incoming agent flow and $\beta$ is the flow on the escalator. Only with this condition it is guaranteed that the escalator is operated in a mode that corresponds to its capacity.

## Model of pedestrians

To account for dynamical factors of the passenger dynamics on escalators, several rule-based models have been developed in the past [22–25]. In this contribution, we do not use one of the existing models but extend the collision-free speed model [26] which, compared to second-order models [27, 28], is characterized by a reduced computational effort and a faster run time of the simulations. The model is minimum in the sense that it is defined by only a small set of parameters. It is also particularly well suited for describing the passenger dynamics on escalators, as will be shown below.

In the simulations, we use circular-shaped agents with a projected diameter of $d = 0.4$ m and a corresponding occupation area of $A \approx 0.13$ m$^2$. According to [10], this space requirement corresponds to the one of an adult wearing thick clothing. In principle, circular-shaped agents can cause gridlocks in the presence of narrow bottlenecks. In the considered simulations, however, all bottlenecks are considerably wider than $w = 0.4$ m and the problem does not occur. In addition, each agent is assigned an individual desired velocity which it tries to reach on the plain. The absolute values of the velocities are provided in the form of a normal distribution for which we use $v_{0,horiz.} = 1.3$ ms$^{-1}$ as expected value and $\sigma = 0.26$ ms$^{-1}$ as standard deviation which is in accordance with [9]. Moreover, we assume the actual speed of an agent to be proportional to the available free space [26]:

$$v(s, T) = \min\left\{v_{0,horiz.}, \max\left\{0, \frac{s - d}{T}\right\}\right\}, \quad T > 0, \tag{1}$$

where $s$ is the smallest distance to the neighbors in the motion direction. The variable $T$ corresponds to the time gap between successive pedestrians in single-file motion [26]. As with the desired velocities, individual $T$-values can be assigned to the agents in form of a normal

distribution. In order to study the impact of $T$ on the numerical results, however, in each of the simulations we use one $T$-value for all agents. This value is then (possibly) varied in successive simulations. A typical value is $T = 0.25$s.

Finally, neighbors and walls not only influence the speed of an agent but also its movement direction. In this work, we use default values for the model parameters as suggested in [21,26,29]: $a_{\text{agent}} = a_{\text{wall}} = 5$, $D_{\text{agent}} = 0.1$m, and $D_{\text{wall}} = 0.02$ m. Here, $a_{\text{agent}}$ and $a_{\text{wall}}$ are dimensionless repulsion coefficients while $D_{\text{agent}}$ and $D_{\text{wall}}$ are repulsion distance thresholds between pedestrians and with the geometry, respectively.

## Model of escalators

So far, we have introduced rules that govern the movement of pedestrians in 2D space in general. However, unlike static geometrical components like corridors and bottlenecks, escalators have their own dynamics and move upwards or downwards with a specific conveyor speed. In *JuPedSim*, an upward or downward escalator is implemented by specifying an inclined area of the geometry in which agents mainly move at the escalator speed. As the desired speed of a walking agent on the plain, $v_{0,\text{horiz.}}$, is typically different from the speed of the escalator, $v_{\text{esc}}$, it becomes necessary to moderate between the two quantities. In *JuPedSim* this is done by means of a smooth adaptation function that avoids any discontinuities [21]. The actual position-dependent speed of the agent is then given by

$$v_0(x) = v_{0,\text{horiz.}}(1 - f(x)) + v_{\text{esc}} \cdot f(x), \tag{2a}$$

$$\text{with } f(x) = \tanh(cx^2) \cdot \tanh\left(c(x - l_{\text{esc}})^2\right), \tag{2b}$$

where $c$ is a parameter to adjust the adaptation function. To take the speed adaptation into account, the replacement $v_{0,\text{horiz.}} \rightarrow v_0$ has to be made in Eq (1).

Fig 3 shows the development of the actual speed as one single agent passes the same escalator for two different values of the parameter $c$ and three common escalator speeds, respectively. In each of the cases, the agent decelerates beginning at $x = 0$ m, adapts to the speed of

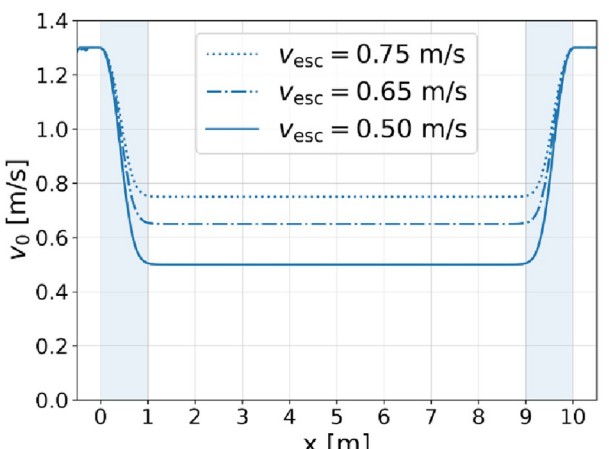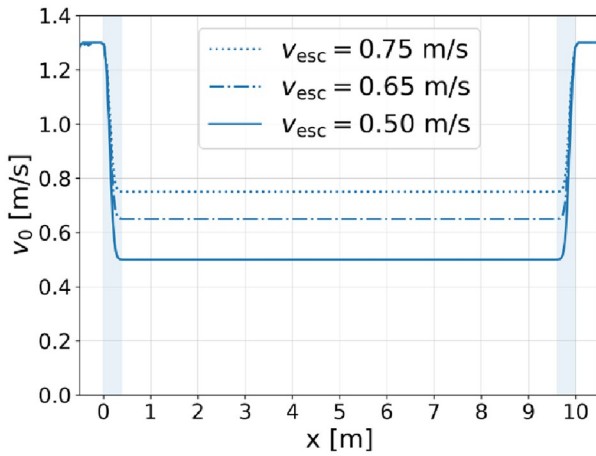

**Fig 3. Speed of a single agent, depending on the position on the escalator.** Shown are simulation results obtained for $c = 50$ (left) and $c = 500$ (right), and for the common escalator speeds $v_{\text{esc}} \in \{0.50, 0.65, 0.75\}$ ms$^{-1}$, respectively. The desired speed of free walking on the plain is here $v_{0,\text{horiz.}} = 1.3$ ms$^{-1}$. The shaded areas indicate regions where the agent speed is adapting to the one of the respective escalator. The corresponding adaptation lengths are approximately 1.0 m (left panel) and 0.4 m (right panel).

**Table 2. Required length of the horizontal flights at the entrance and exit of an escalator according to norm EN 115–1 [7].**

| escalator speed | minimum length |
|---:|---|
| $v_{esc} \leq 0.50 \text{ ms}^{-1}$ | 0.8 m ($\approx$ 2 steps) |
| $0.50 \text{ ms}^{-1} < v_{esc} \leq 0.65 \text{ ms}^{-1}$ | 1.2 m ($\approx$ 3 steps) |
| $0.65 \text{ ms}^{-1} < v_{esc} \leq 0.75 \text{ ms}^{-1}$ | 1.6 m ($\approx$ 4 steps) |

The values are valid for heights $\Delta h \leq 6$ m.

the escalator, holds it, and accelerates at $x > 9$m to reach again the desired speed of free walking on the plain.

On a real-life escalator, the adaptation process is supposed to take place in the horizontal flights at the entrance and the exit areas. According to norm EN 115–1:2017 [7], the minimum length required in movement direction of these regions depends on the conveyor height, slope, and speed of the escalator, see e.g. Table 2. Comparing the requirements with Fig 3, it follows that the deceleration and acceleration of the agent can be adjusted such that it takes place fast enough. More precisely: for $c = 50$, the adaptation length is approximately 1.0 m which, according to Table 2, would be too long for escalator speeds $v_{esc} \leq 0.5 \text{ ms}^{-1}$. Setting $c = 500$, however, the adaptation length can be shortened to approximately 0.4 m which is in accordance with the requirements of norm EN 115–1.

## Simulation of multiple agents

As mentioned before, in *JuPedSim* escalators are modeled such that each agent is assigned an individual *desired* speed which it tries to attain passing the facility. When considering a single agent, this method is definitely sufficient to obtain realistic results since a single agent can choose its speed almost completely freely. In crowded situations, however, interactions among the agents limit this freedom and, accordingly, the simulated speed is likely to be below the desired one. This is in contrast to the situation at an escalator since here the conveyor speed constitutes a lower bound for the speed of the agents located on it.

In order to estimate whether it is possible to model realistic behavior on escalators with the concept of a desired speed, we consider the following setup: We use the geometry of Fig 2 and simulate 1.000 agents passing successively an escalator that is operated at a speed of $v_{esc} = 0.5$ ms$^{-1}$. The continuous incoming agent flow of $\alpha = 3$s$^{-1}$ is chosen such that it lies above the capacity of the escalator, as will be shown below. The corresponding simulation results are summarized in Fig 4.

As expected, all agents are adapting their speed to match the speed of the escalator. Since $\alpha$ is larger than the escalator capacity, a congestion forms in front of the facility leading to stronger interactions among the agents and therefore to a reduction of the individual speeds in the region $x < 0$ m. As a consequence, most of the agents have to accelerate instead of decelerate when entering the escalator which is reflected by the fact that Fig 4 is asymmetrical around $x = 5$ m (downward shift of the l.h.s.). Compared to the right diagram in Fig 3, also the adaptation length is slightly extended at the entrance of the escalator. However, it is, for the vast majority of the agents, still compatible with the requirements of Table 2. Outside the adaptation phases (in the region $1$m $< x < 9$m) all agents are moving with approximately the same speed which corresponds to the case of standing agents on the escalator.

Finally, the results in Fig 4 show that it is indeed possible to model realistic behavior on escalators with the concept of a desired speed, even for the case that the incoming agent flow is

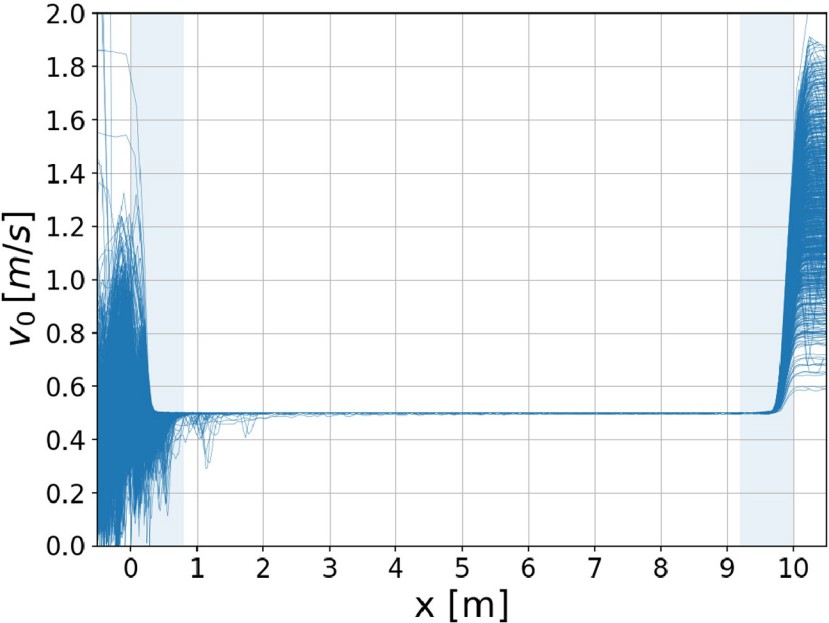

**Fig 4. Individual speeds of 1.000 agents, depending on the position on the escalator.** The results are obtained using $c = 500$, $v_{\text{esc}} = 0.5\,\text{ms}^{-1}$, and $w = 1.0$ m. The shaded areas indicated the required length of the horizontal flights according to Table 2.

significantly larger than the handling capacity of the facility. The agreement between simulation results and simulation goal is particularly striking for escalator speeds $v_{\text{esc}} \leq 0.5\,\text{ms}^{-1}$.

## Passenger dynamics of escalators

In the following, we examine different quantities with simulations that are difficult to access through experiments and field studies. More precisely, we consider situations that involve the operation of escalators *above* their respective capacity, meaning that the incoming agent flow $\alpha$ is larger than the maximum flow $\beta$ that can be transported away by the escalator. In real-life situations, these situation are either too dangerous for the involved people or the necessary boundary conditions can only be ensured for a short amount of time. In contrast, the following results (and underlying simulations) do not suffer from these restrictions. It is therefore possible to realize and investigate steady-states of almost any temporal extension.

### Spatial separation of agents on the escalator

One particular feature of an escalator with standing agents is that the distances between any agent and its direct neighbors do not change, even though the overall agent flow is non-vanishing. To verify if the model introduced in the previous section recovers this feature, in each simulation we label the $n$ agents currently located on the escalator according to their position in movement direction (in our case $x$-direction) and consider all $(n-1)$ spatial distances between adjacent agents. The average along the $x$- and the $y$-axis is then given by

$$\Delta x(t_0) = \frac{1}{n(t_0) - 1} \sum_{i=1}^{n(t_0)-1} \left| x_{i+1}(t_0) - x_i(t_0) \right|, \tag{3a}$$

$$\Delta y(t_0) = \frac{1}{n(t_0) - 1} \sum_{i=1}^{n(t_0)-1} \left| y_{i+1}(t_0) - y_i(t_0) \right|, \tag{3b}$$

where $x_j$ is the absolute $x$-position of agent $j$ and similar for the $y$-direction. In a similar way, it is possible to determine the total distance $s$:

$$s(t_0) = \frac{1}{n(t_0) - 1} \sum_{i=1}^{n(t_0)-1} \sqrt{\left| x_{i+1}(t_0) - x_i(t_0) \right|^2 + \left| y_{i+1}(t_0) - y_i(t_0) \right|^2}. \tag{3c}$$

Simulation results of the different distances are shown in Fig 5.

As can be seen, after an initial phase where the first agents are arriving at the escalator, the average distance in movement direction is dropping and a steady-state with constant spatial separations establishes for each of the considered escalators. The only quantity in Fig 5 that shows a significant speed dependence for *both* escalator widths is $\Delta x$, or more specifically, $\overline{\Delta x}$

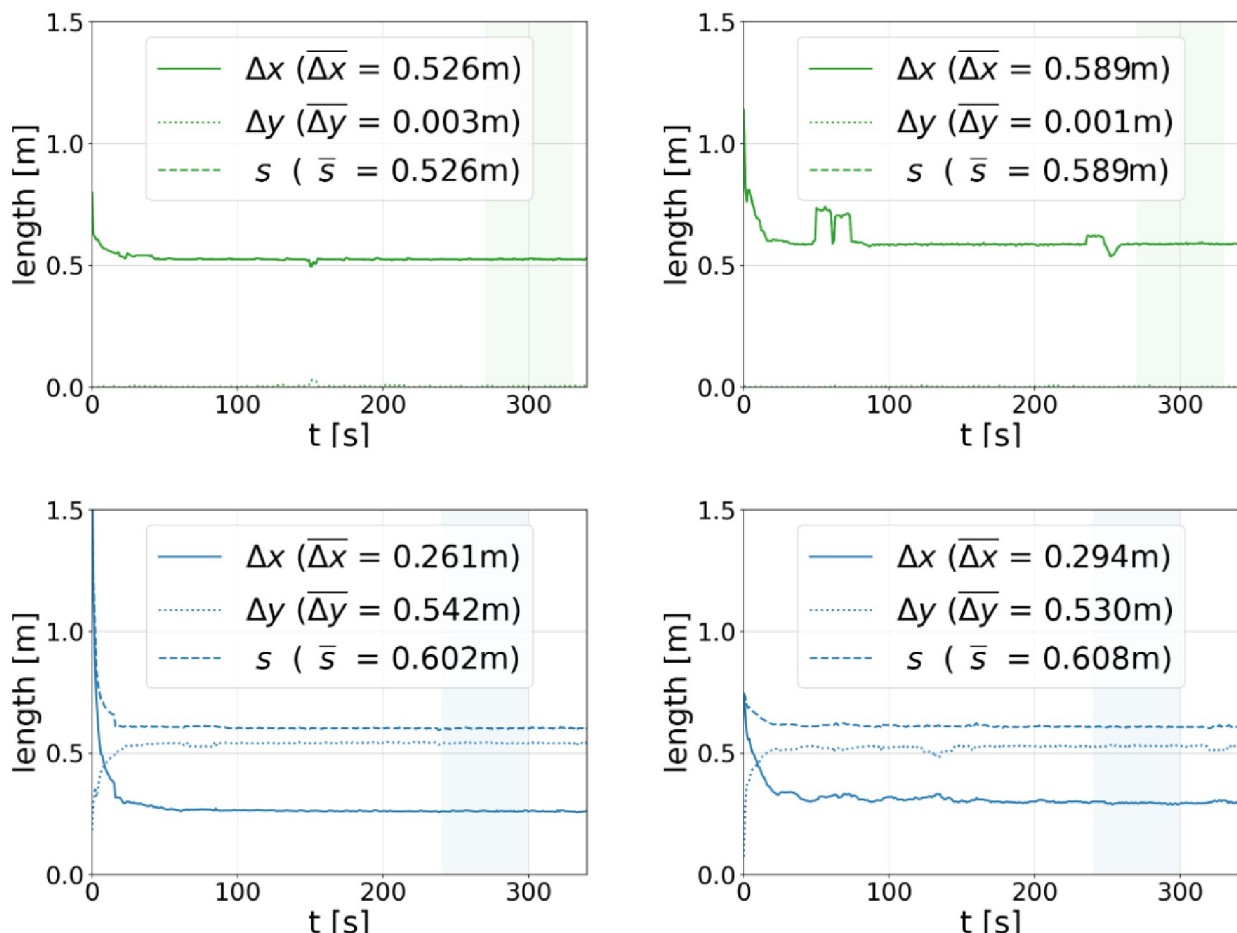

**Fig 5. Simulation results for spatial distances between the agents.** Shown are results for the clear widths $w = 0.6$ m (first row) and $w = 1.0$ m (second row), as well as for the escalator speeds $v_{esc} = 0.5$ ms$^{-1}$ (left) and $v_{esc} = 0.75$ ms$^{-1}$ (right), respectively. The average values of the distances are obtained from steady-state regions of 60s length (shaded areas) and are indicated by a bar. Since $\Delta y$ almost vanishes in the upper two diagrams, the total distance $s$ is approximately the same as $\Delta x$.

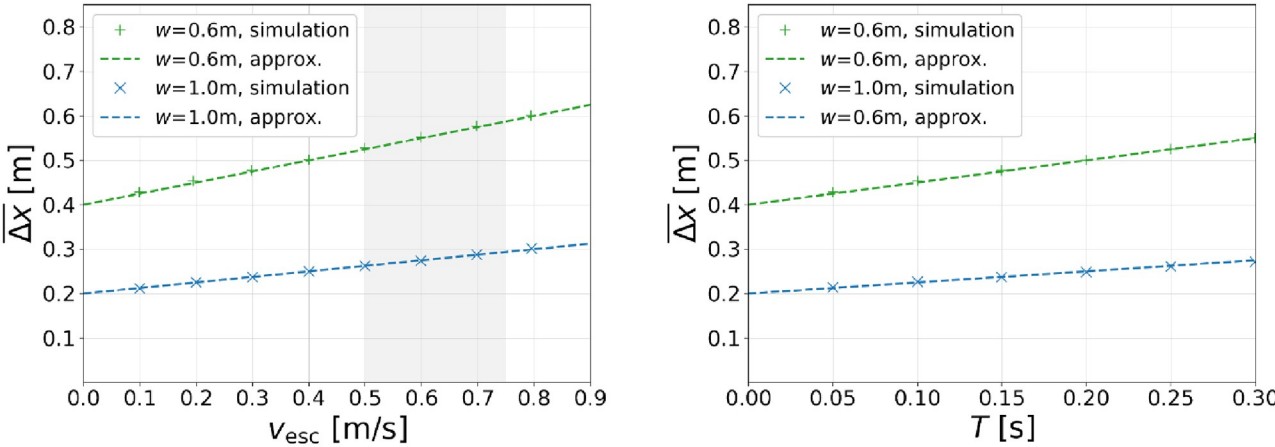

**Fig 6. Minimum spatial separations in movement direction depending on the escalator speed (left) and the model parameter *T* (right).** The simulation results are obtained by measuring $\overline{\Delta x}$ for the escalator widths $w = 0.6$ m and $w = 1.0$ m, respectively. Model parameters are fixed by $T = 0.25$s (left) and $v_{esc} = 0.5$ ms$^{-1}$ (right). The approximations follow from Eq (6) using $d_{step} = 0.4$ m, $\mathcal{O}_0 = 2$ for $w = 1.0$ m, and $\mathcal{O}_0 = 1$ for $w = 0.6$ m. The shaded area indicates the range of typical escalator speeds.

which is the average of $\Delta x$ over all considered frames in the steady-state (shaded areas). To further study this dependence, we consider a wider range of escalator speeds and determine the average of the *minimum* spatial separations in *x*-direction as before. The corresponding simulation results are summarized in the left diagram of Fig 6. For both escalator widths and over the entire considered speed range, $\overline{\Delta x}$ depends *linearly* on $v_{esc}$. The obtained simulation results can therefore be fitted by a linear regression for which we obtain

$$\text{fit for } w = 0.6 \text{ m}: \quad \overline{\Delta x}(v_{esc}) = 0.404 \text{ m} + 0.248 \text{ s} \cdot v_{esc}, \quad T = 0.25 \text{ s}, \tag{4a}$$

$$\text{fit for } w = 1.0 \text{ m}: \quad \overline{\Delta x}(v_{esc}) = 0.199 \text{ m} + 0.127 \text{ s} \cdot v_{esc}, \quad T = 0.25 \text{ s}. \tag{4b}$$

In order to check the impact of the model parameter *T*, we vary the *T*-value for the fixed escalator speed $v_{esc} = 0.5$ ms$^{-1}$ with corresponding simulation results shown in the right diagram of Fig 6.

As before, the simulation results show a linear parameter dependence and can be fitted accordingly:

$$\text{fit for } w = 0.6 \text{ m}: \quad \overline{\Delta x}(T) = 0.405 \text{ m} + 0.482 \text{ ms}^{-1} \cdot T, \quad v_{esc} = 0.5 \text{ ms}^{-1}, \tag{5a}$$

$$\text{fit for } w = 1.0 \text{ m}: \quad \overline{\Delta x}(T) = 0.203 \text{ m} + 0.229 \text{ ms}^{-1} \cdot T, \quad v_{esc} = 0.5 \text{ ms}^{-1}. \tag{5b}$$

The results in Eqs (4) and (5) can be explained as follows: The typical step depth of an escalator is $d_{step} = 0.4$ m [3–5]. On the narrow escalator with a clear width of $w = 0.6$ m, only one person can stand on each step simultaneously, resulting in a minimum possible *x*-distance between the agents of $\min(\overline{\Delta x}) = 0.4$ m. In contrast, for escalators with a width of $w = 1.0$ m, two persons can stand next to each other on each step, resulting in a minimum *x*-distance $\min(\overline{\Delta x}) \approx 0.2$ m.

Introducing the step depth of an escalator, $d_{\text{step}}$, as a new variable, Eqs (4) and (5) can therefore be approximated and rewritten in a unified form as

$$\overline{\Delta x}(w, v_{\text{esc}}) \simeq \frac{1}{\mathcal{O}_0(w)}\left(d_{\text{step}} + T \cdot v_{\text{esc}}\right), \quad T, v_{\text{esc}} > 0. \tag{6}$$

In this expression, the whole width dependence is factored out in terms of the parameter $\mathcal{O}_0(w)$ which is given by $\mathcal{O}_0 = 1$ for $w = 0.6$ m and $\mathcal{O}_0 = 2$ for $w = 1.0$ m. One particular feature of Eq (6) is that it explicitly contains the model parameter $T$ which governs the inverse slope of the speed function, see for instance Eq (1). In the present case of an escalator with congestion upstream, $T$ acts in the role of a time gap as most of the agents have to accelerate when entering the escalator which in turn is a consequence of the congestion in front of the facility (the incoming agent flow $\alpha$ is larger than the escalator capacity, i.e. $C_{\text{esc}} = \max(\beta) < \alpha$). The acceleration of each agent can only be initiated in case there is free space available at the entrance of the escalator. For non-vanishing values of $T$ and $v_{\text{esc}}$, this fact causes additional spatial gaps in movement direction between the standing agents since the escalator is constantly moving on during the entering process. Conversely, in the (hypothetical) case that $T$ vanishes, the spatial separation in movement direction coincides with the step depth $d_{\text{step}}$. Accordingly, the term $T \cdot v_{\text{esc}}$ can be regarded as a 'reaction distance' which is an inevitable consequence of the fact that $T$ is non-vanishing. Moreover, apart from the global prefactor $\mathcal{O}_0$, the reaction distance appears without any non-trivial coefficient and exponent. In this way, the 'unphysical' model parameter $T$ directly translates into a 'physical' observable that can be measured.

We would like to stress that, apart from $T$ and $v_{\text{esc}}$, the simulation results have been obtained by using *identical* values for the other model parameters. Although individual escalator steps are not implemented in the simulations, reasonable results for the facility parameter $d_{\text{step}}$ are obtained as a consequence of the used parameters, in particular the projected diameter of the agents. We have also checked explicitly that the obtained results are independent of the model parameter $c$, introduced in Eq (2) to govern the speed adaptation of the agents. Overall, the deviation between the different simulation results in Fig 6 and the approximation in Eq (6) and is less than one per cent in the considered parameter range.

## Step occupancy and density on the escalator

In contrast to the previous considerations, the spatial distances in movement direction between standing agents are discretized on a real-life escalator due to the presence of individual (moving) steps. In other words, the horizontal length of an escalator, $l_{\text{esc}}$, is divided into $n_{\text{steps}}$ steps of depth $d_{\text{step}}$:

$$l_{\text{esc}} = n_{\text{steps}} \cdot d_{\text{step}}. \tag{7}$$

As mentioned before, the actual number of persons that can simultaneously stand on the same step depends on the clear width $w$ of the escalator. To quantify this fact, we introduce the so-called *step occupancy* $\mathcal{O}_i(t_0)$ which counts the absolute number of standing agents on step $i$ at time $t_0$. Considering persons with a shoulder width of $d = 0.4$ m as before, it is obvious that the following step occupancies can be realized:

$$0.4 \text{ m} \leq w < 0.8 \text{ m}: \quad \mathcal{O}_i(t_0) \in \{0, 1\}, \tag{8a}$$

$$0.8 \text{ m} \leq w < 1.2 \text{ m}: \quad \mathcal{O}_i(t_0) \in \{0, 1, 2\}. \tag{8b}$$

The *average* step occupancy of an escalator can then be computed by simply taking the average of all steps, i.e.

$$\overline{\mathcal{O}}_{n_{\text{steps}}}(t_0) = \frac{1}{n_{\text{steps}}} \sum_{i=1}^{n_{\text{steps}}} \mathcal{O}_i(t_0) \equiv \frac{1}{n_{\text{steps}}} n_{\text{esc}}(t_0)(7)(7) \frac{d_{\text{step}}}{l_{\text{esc}}} n_{\text{esc}}(t_0),$$ (9)

where $n_{\text{esc}}$ is the number of standing agents on the escalator.

In Eq (9), the step occupancy is defined in a *macroscopic* way by considering the whole escalator at a time and by measuring the absolute number of agents currently located on it. Apart from this global definition, it is also possible to consider only the *occupied* area of an escalator by using the results of the spatial distances from the previous section. For example, on an escalator with a step depth of $d_{\text{step}} = 0.4$ m the average distance between the agents in movement direction is $\overline{\Delta x} = 0.4$ m if each step is occupied by one person. If only every second step is occupied, the average distance between the agents is $\overline{\Delta x} = 0.8$ m etc. This relationship between the (macroscopic) occupancy and the (microscopic) spatial separation can be written as

$$\overline{\mathcal{O}}_{\Delta x}(t_0) = \frac{d_{\text{step}}}{\overline{\Delta x}(t_0)}.$$ (10)

Since both quantities on the r.h.s. are known from Eq (6), we can write the occupancy as

$$\overline{\mathcal{O}}_{\Delta x} \simeq \frac{\mathcal{O}_0(w) \cdot d_{\text{step}}}{d_{\text{step}} + T \cdot v_{\text{esc}}}, \quad T, v_{\text{esc}} > 0.$$ (11)

Simulation results based on this definition of the step occupancy are shown in the left diagram of Fig 7.

The simulation results show that the average step occupancy

- decreases with increasing escalator speed; the decrease is (slightly) non-linear,

- stays below $\mathcal{O} = 1$ for the escalator width $w = 0.6$ m which is compatible with the fact that at most one person can simultaneously occupy one step, see Eq (8a),

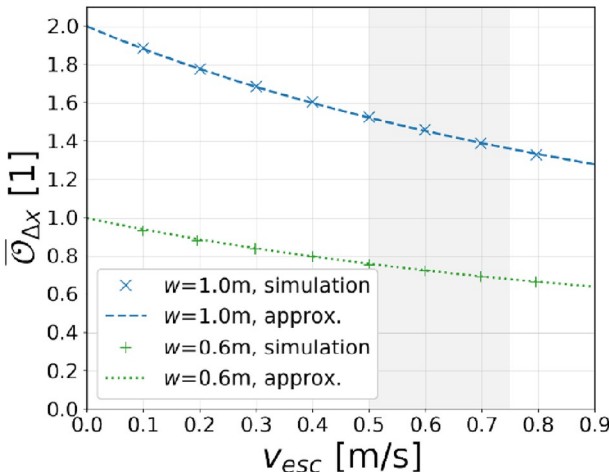
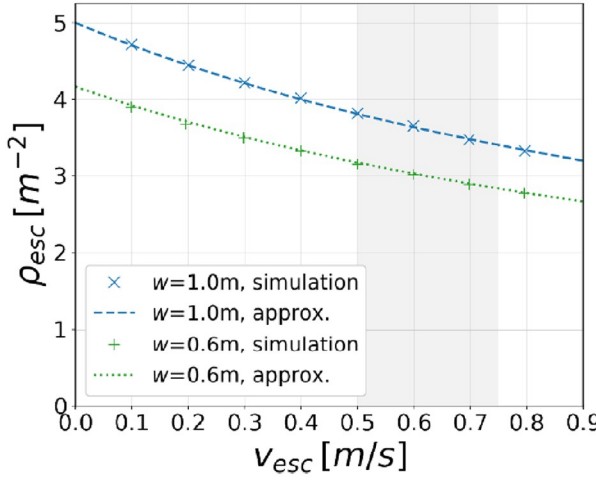

**Fig 7. Speed dependence of the step occupancy.** The simulation results are obtained from Eq (10) by using results of the spatial separation in movement direction from the previous section. The respective approximations follow from Eq (11) using $T = 0.25$s, $d_{\text{step}} = 0.4$ m, $\mathcal{O} = 2$ for $w = 1.0$ m, and $\mathcal{O} = 1$ for $w = 0.6$ m.

- stays below $\mathcal{O} = 2$ for the escalator width $w = 1.0$ m which is compatible with the fact that at most two persons can simultaneously occupy one step, see Eq (8b).

As mentioned before, at a real-life escalator distances between standing agents in movement direction are discretized due to the presence of individual steps. Larger spatial distances as a consequence of a finite time gap between entering passengers can therefore only be realized in the form of an under-occupation of individual steps. Even though individual steps are not explicitly considered in the presented space-continuous model, the obtained simulation results as well as Eq (11) recover this feature. In this way, the left diagram in Fig 7 also explains observations already formulated in [20]:

'Numerous observations have shown that 100 per cent step utilization is never obtained, even with the heaviest traffic pressure and use by the most knowledgeable and agile pedestrians, including commuters.'

In addition, the meaning of the previously introduced parameter $\mathcal{O}$ becomes clear. It is the *maximum* occupancy that can only be realized in the hypothetical limits $T \to 0$ and $v_{esc} \to 0$:

$$\lim_{T \to 0} \overline{\mathcal{O}}_{\Delta x} = \lim_{v_{esc} \to 0} \overline{\mathcal{O}}_{\Delta x} = \mathcal{O}_0(w). \tag{12}$$

In all considered simulations, the escalators are set up in an operational mode that is above their respective capacity. Eqs (9) and (10) should therefore yield similar results for the steady-states since the escalators are continuously fully occupied. We have checked explicitly that the relation $\overline{\mathcal{O}}_{n_{steps}} \approx \overline{\mathcal{O}}_{\Delta x}$ indeed holds. Moreover, the maximum deviation between the individual simulation results and the approximation in Eq (11) is at most one per cent.

Finally, we would like to mention that, for a given escalator, the quantities *occupancy* and *density* are closely related but not identical. For instance, the density on an escalator can be obtained by dividing the total number of agents located on the escalator, $n_{esc}$, by the projected area of the escalator, $A_{esc} = w \cdot l_{esc}$. Using Eqs (9) and (10), we therefore write the relation between density and occupancy as

$$\rho_{esc} = \frac{n_{esc}}{A_{esc}} = \frac{n_{esc}}{w \cdot l_{esc}} \stackrel{(9)}{=} \frac{\overline{\mathcal{O}}_{n_{steps}} \cdot l_{esc}}{d_{step} \cdot w \cdot l_{esc}} = \frac{\overline{\mathcal{O}}_{n_{steps}}}{d_{step} \cdot w}, \tag{13a}$$

$$\approx \frac{\overline{\mathcal{O}}_{\Delta x}}{d_{step} \cdot w} \stackrel{(10)}{=} \frac{d_{step}}{d_{step} \cdot \overline{\Delta x} \cdot w} = \frac{1}{\overline{\Delta x} \cdot w}. \tag{13b}$$

In the second line, we made use of the relation $\overline{\mathcal{O}}_{n_{steps}} \approx \overline{\mathcal{O}}_{\Delta x}$ which is valid for the case that the escalator is fully occupied. Simulation results for the density on escalators are shown in the right diagram of Fig 7.

## Escalator capacity

One key measure of an escalator is its capacity which, for a given time interval $\Delta t = t_1 - t_0$, is the maximum number of persons that can be transported away by the facility, i.e.

$$C_{esc}(w, v_{esc}) = \max\{\beta(w, v_{esc})\} = \max\left\{\frac{n(t_1) - n(t_0)}{\Delta t}\right\}. \tag{14}$$

According to this definition, the determination of the capacity consists of two independent

measurements as one has to determine the agent number (at least) at two different times $t_1$ and $t_0$. At an escalator, the situation is slightly different since here the conveyor speed is continuously constant. In order to obtain the capacity, it is therefore sufficient to measure only one quantity at a fixed time. In general, there are two options:

(a) **measurement of the agent number**:

The starting point of this approach is the hydro-dynamical equation $\beta = \rho_{esc} \cdot w \cdot v_{esc}$ together with the definition of the capacity in Eq (14). Using Eq (13a), the capacity is given by

$$C_{esc}(w, v_{esc}) = \frac{\max\{n_{esc}(w, v_{esc})\}}{l_{esc}} \cdot v_{esc}, \quad \alpha > C_{esc}. \tag{15a}$$

(b) **measurement of the spatial separation in movement direction**:

As in approach (a), the starting point is the hydro-dynamical equation. However, using Eq (13b), the density is expressed in terms of the spatial separation in movement direction, i.e.

$$C_{esc}(w, v_{esc}) = \frac{1}{\min\{\overline{\Delta x}(w, v_{esc})\}} \cdot v_{esc}, \quad \alpha > C_{esc}. \tag{15b}$$

Conceptually, approach (a) is based on the measurement of agent numbers which can be carried out comparatively easily. The obtained results are, however, very sensitive to small fluctuations of the agent numbers. For example, for an escalator with a clear width of $w = 0.6$ m and a length of $l_{esc} = 10$ m, a step depth of $d_{step} = 0.4$ m results in 25 steps, each with a maximum occupancy of $\max(\mathcal{O}_i) = 1$. A maximum of $n_{esc} = 25$ agents can therefore simultaneously be located on the escalator. In this case, each agent contributes at least 4% to the result in Eq (15a). For shorter escalators, the rate is even higher. In contrast, the measurement of $\Delta x$ in approach (b) is mainly limited by the spatial resolution of the simulations, resulting in overall smaller uncertainties. On a commercial escalator, the smallest possible value of the spatial separation in movement direction is $\overline{\Delta x} \approx 0.2$ m, see for instance Fig 6. A spatial resolution of 0.001m, as used in the simulations, then corresponds to a precision of 0.5%. In the following, we therefore use Eq (15b) to measure the speed-dependent capacity of the escalators. In doing so, we check explicitly that the obtained results are compatible with the other approach within the measurement uncertainties.

In Eq (15b), the capacity is expressed in terms of spatial separations in movement direction. Using Eq (6), we can therefore write

$$C_{esc}(w, v_{esc}, T) \simeq \mathcal{O}_0(w) \cdot \frac{v_{esc}}{d_{step} + T \cdot v_{esc}}, \quad T, v_{esc} > 0, \tag{16}$$

where again $\mathcal{O}_0 = 2$ for $w = 1.0$ m and $\mathcal{O}_0 = 1$ for $w = 0.6$ m. This *non-linear* function for the speed-dependent capacity of an escalator is one of the most important findings in this contribution. To the knowledge of the authors, it is for the first time that such a formula is given in an explicit analytical form only including physical parameters. A graphical representation of the capacity formula is shown in Fig 8.

Regarding large escalator speeds, i.e. $v_{esc} \rightarrow \infty$, Eq (16) has no local maximum, but is bounded above by

$$\lim_{v_{esc} \rightarrow \infty} C_{esc}(w, v_{esc}, T) = \frac{\mathcal{O}_0(w)}{T}. \tag{17}$$

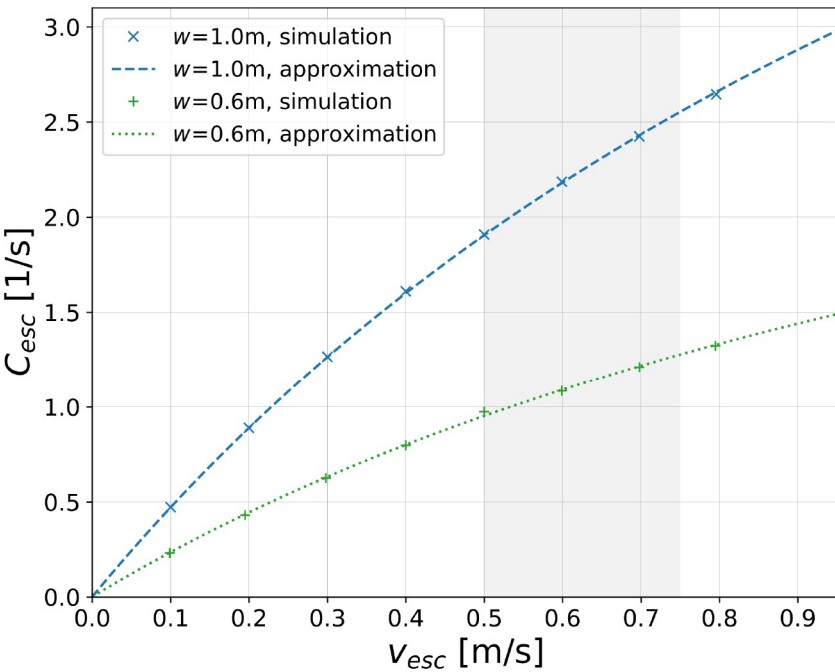

**Fig 8. Non-linear speed-dependence of the escalator capacity.** Shown are simulation results for the escalator widths $w = 1.0$ m (blue markers) and $w = 0.6$ m (green markers). Approximations are obtained by using Eq (16) with $d_{step} =$ 0.4 m and $T = 0.25$ s. The shaded area indicates the range of typical escalator speeds.

In other words, the capacity of an escalator is not only limited by the clear width of the facility, but also by the value of the parameter $T$. In the present case of an escalator with congestion upstream, the parameter can be regarded as a reaction time of the passengers entering the facility, see for instance Eq (6) and the discussion below that formula. Only in the limit of a vanishing time gap $T$, the capacity can get in principle arbitrarily large. In Eq (16), this hypothetical (!) case then corresponds to a linear rise of $C_{esc}$ with increasing $v_{esc}$.

The fact that Eq (16) explicitly contains the model parameter $T$ is a strong indication that the underlying collision-free speed model [26] is very well suited for describing the passenger dynamics of escalators, at least for the considered escalator widths. Moreover, we would like to stress that $T$ is the only 'free' parameter in Eq (16) that can be used to adjust the simulations. All other parameters are properties of the escalator itself and are therefore fixed for a given scenario. In this respect, we have checked explicitly that the obtained capacity values are independent of parameter $c$ which is related to the speed adaptation of the agents, see Eq (2). The $T$-dependence of the capacity in the range $0.15s \leq T \leq 0.3s$ is shown in Fig 9.

To further verify the obtained results, we compare them with field studies summarized in Table 1 and with data provided by escalator manufacturers. When comparing Eq (16) with the literature, the following points should be noted:

- Results of field studies are only available for the commonly used escalator width $w = 1.0$ m. In Fig 9, therefore only results for $w = 1.0$ m are shown. All following statements are, however, also valid for $w = 0.6$ m.

- Eq (16) describes the capacity of an escalator which is the *maximum* agent flow that can be realized over a certain period of time. Since the necessary boundary conditions are difficult to come across in real-life situations, observed results of field studies should lie *on or below*

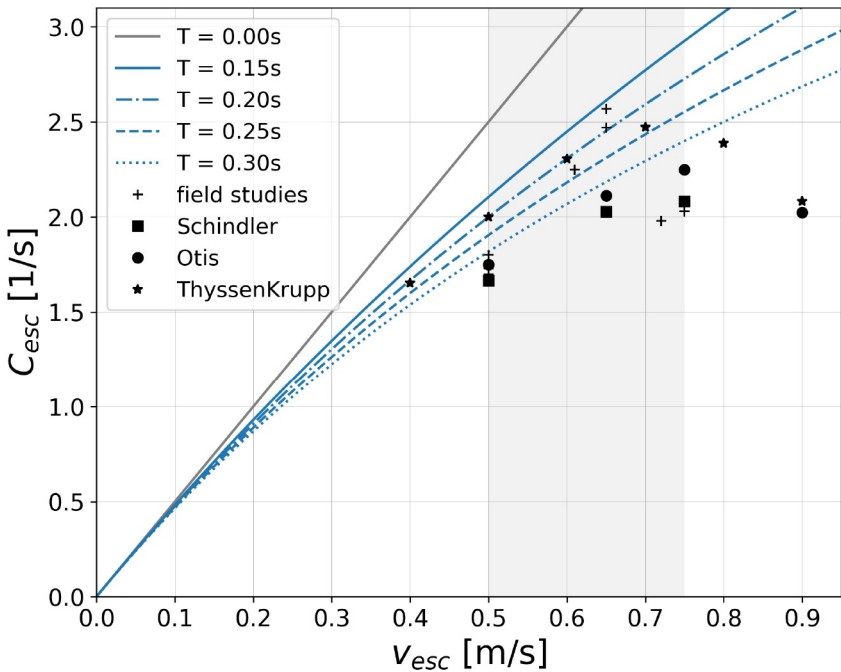

**Fig 9. Speed-dependent capacity of an escalator for different values of the time gap $T$.** The diagram compares Eq (16) for different $T$-values with maximum observed flows summarized in Table 1 and 'practical' capacities provided by manufacturers [3–5]. The escalator parameters are fixed by $d_{step}$ = 0.4 m and $w$ = 1 m ($\mathcal{O}_0$ = 2). By definition, the capacity should be as big as or larger than observed flows.

the respective curve in Fig 9. Depending on which entries in Table 1 are considered valid, $T$ can be chosen such that this is indeed the case.

- The minimum value of the time gap to explain *all* results in Table 1 is $T$ = 0.15s. Using this value, Eq (16) yields a (speed-dependent) escalator capacity that is slightly larger than all observed flows, including the ones obtained for escalator speeds $v_{esc} \geq 0.65$ ms$^{-1}$, see Fig 9.

- Considering the most commonly used escalator speed, $v_{esc}$ = 0.5 ms$^{-1}$, a minimum time gap of $T$ = 0.2s is sufficient to explain all results in Table 1 and data provided by the manufacturers.

- A time gap of $T$ = 0.35s is too long to be compatible with observed flows of [12, 14–16] and data provided by [3–5]. In order to explain these experimental data with Eq (16), therefore smaller $T$-values have to be used, see the above bullet points.

Summarizing, the comparison of Eq (16) with available literature on escalator capacities shows that the applied setup as well as the introduced escalator model yield reliable and sound predictions. As one of the most important findings, we deduce a minimum additional time gap between entering passengers in the range of $T = (0.15 − 0.30)$s which is perfectly compatible with knowledge of social psychology on the minimum human reaction time, see e.g. [30–34].

This fact provides further evidence that, at least for the case at hand (escalator with congestion upstream), $T$ can be interpreted as human reaction time. Moreover, the finite value of the additional time gap is the fundamental reason for a capacity reduction of the escalator. If the escalator is operated at the most commonly used speed $v_{esc}$ = 0.5 ms$^{-1}$, for instance, the capacity reduction amounts up to 27% compared to the (hypothetical) case $T$ = 0.

So far, in the simulations we consider a constant model parameter $T$ which possibly may have an individual value for each simulated agent, but which is definitely independent of the escalator speed $v_{esc}$. The case that the time gap itself depends on the escalator speed is discussed in the S1 Appendix.

## Conclusions and outlook

The passenger dynamics of a pedestrian facility depends, in general, on properties of the facility (geometry, clear width, inclination angle, etc.) and on characteristics of the passengers (size, shape, desired speed, etc.). Typically, properties of the facility are known or can be assessed relatively easily. Agent characteristics, in contrast, are individual and can in general only be known approximately. Moreover, the multitude of the latter usually makes it difficult to identify structural dependences as the quantities are numerous and even interdependent. In this regard, escalators with standing agents located on it constitute an exception in the sense that many agents properties have no significant impact on the passenger dynamics of the facility.

Using the model presented in this paper, we find in Eq (16) that the only agent-related property that affects the escalator capacity is the parameter $T$ of the underlying collision-free speed model. In the case at hand, this parameter is related to the entering process of the agents and quantifies an additional time gap between two entering passengers. Escalators are therefore a practical tool to study in particular this quantity. Regarding the $T$-dependence, two cases can be distinguished:

(I) The (hypothetical) case $T = 0$ (no passenger reaction time) results in a *linear* speed dependence of the capacity. In the literature, this case is often referred to as 'theoretical' capacity. Considering Eq (16), it becomes clear that this case is an unrealistic simplification of the prevailing circumstances which therefore does not coincide with empirical observations.

(II) In contrast, the case $T > 0$ (positive passenger reaction time) results in a *non-linear* speed dependence of the capacity. The deviation from linearity is particularly evident in the range of large escalator speeds $v_{esc} > 0.5 \text{ ms}^{-1}$. The reason is that during each additional time gap, the escalator is constantly moving on, resulting in larger spatial gaps between the agents standing on the escalator. Of course, this effect is enhanced with larger $v_{esc}$. Even for the moderate and most commonly used escalator speed $v_{esc} = 0.5 \text{ ms}^{-1}$, a value of $T = 0.25$s leads to a capacity reduction of approximately one fourth compared to the hypothetical case $T = 0$.

The capacity formula developed in this work is surprisingly simple but agrees remarkably well with available literature on escalator capacities for $T$-values in the range $T = (0.15 - 0.30)$s. Furthermore, obtained results of the additional time gap are in perfect agreement with knowledge of social psychology on the human reaction time. This fact serves as an independent consistency check of both the introduced escalator model and of the applied setup as a whole. In this way, the deduced capacity formula delivers a solid basis for the performance analysis of escalators and contributes to the decision-making of operators to find the optimal conveyor speed. It should be noted that complex passenger behavior like walking on escalators as well as other 'escalator etiquette' is beyond the scope of this work. Also regarding the speed-dependence of the time gap, only first considerations have been conducted in the present contribution, see for instance the S1 Appendix. These aspects as well as their impact on the escalator capacity should be carefully integrated into future work.

## Supporting information

**S1 Appendix. Possible speed dependence of the time gap [3, 5, 19].**
(PDF)

## Author Contributions

**Conceptualization:** Christoph Gnendiger.

**Data curation:** Christoph Gnendiger.

**Formal analysis:** Christoph Gnendiger.

**Investigation:** Christoph Gnendiger.

**Methodology:** Christoph Gnendiger.

**Software:** Christoph Gnendiger, Mohcine Chraibi.

**Validation:** Antoine Tordeux.

**Visualization:** Christoph Gnendiger, Mohcine Chraibi.

**Writing – original draft:** Christoph Gnendiger.

**Writing – review & editing:** Christoph Gnendiger, Mohcine Chraibi, Antoine Tordeux.

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
