## [Decision Letter · Decision Letter 0]

24 Nov 2022

PONE-D-22-26795Come together: A unified description of the escalator capacityPLOS ONE

Dear Dr. Gnendiger,

Thank you for submitting your manuscript to PLOS ONE. After careful consideration, we feel that it has merit but does not fully meet PLOS ONE’s publication criteria as it currently stands. Therefore, we invite you to submit a revised version of the manuscript that addresses the points raised during the review process.

Please review this manuscript according to the comments and questions raised in the reports from the two referees.

We look forward to receiving your revised manuscript.

Kind regards,

José S. Andrade Jr.

Academic Editor

PLOS ONE

Journal Requirements:

"This work was supported by the German Federal Ministry of Education and Research (BMBF) within the framework of the project KapaKrit under the grant number 13N14620. MC and AT acknowledge the Franco-German research project MADRAS funded in France by the Agence Nationale de la Recherche (ANR, French National Research Agency) under the grant number ANR-20-CE92-0033, and in Germany by the Deutsche Forschungsgemeinschaft (DFG, German Research Foundation) under the grant number 446168800."

"This work was supported by the German Federal Ministry of Education and Research (BMBF) within the framework of the project KapaKrit under the grant number 13N14620. MC and AT acknowledge the Franco-German research project MADRAS funded in France by the Agence Nationale de la Recherche (ANR, French National Research Agency) under the grant number ANR-20-CE92-0033, and in Germany by the Deutsche Forschungsgemeinschaft (DFG, German Research Foundation) under the grant number 446168800.

Please include your amended statements within your cover letter; we will change the online submission form on your b

Reviewers' comments:

Reviewer's Responses to Questions

**Comments to the Author**

1. Is the manuscript technically sound, and do the data support the conclusions?

Reviewer #1: Yes

Reviewer #2: Partly

2. Has the statistical analysis been performed appropriately and rigorously? 

Reviewer #1: Yes

Reviewer #2: I Don't Know

3. Have the authors made all data underlying the findings in their manuscript fully available?

Reviewer #1: Yes

Reviewer #2: Yes

4. Is the manuscript presented in an intelligible fashion and written in standard English?

Reviewer #1: Yes

Reviewer #2: Yes

5. Review Comments to the Author

Reviewer #1: The authors studied the dynamics of passengers on escalators through the perspective of numerical simulation. They derived a space-continuous model to describe how agents change from walking on the plain to standing on the escalator. They also deduced a capacity formula that is in agreement with results found in literature. Finally, they claim that the deduced capacity formula allows one to find an optimal conveyor speed simultaneously compatible with safety, comfort, and performance demands.

Overall the manuscript is well written and organized. It brings contributions to the understanding of the dynamics of passengers on escalators, and it seems appropriate for PLOS ONE. However, I have some questions that I would like addressed:

In section "Simulation of escalators", the authors describe some parameters used in the simulations. Precisely, in the last paragraph the authors write "we ensure that the relation α > β holds in all simulations, where α is the incoming agent flow and β is the flow on the escalator." Do the authors believe that the α/β ratio could affect the results? Why or why not?

In section "Speed dependence of the time gap" the authors write "Regarding in particular capacity values stated in [3, 5, 19], however, good agreement is achieved by using A5 = 0.6 s^6m^−5 and Aj = 0 for the other coefficients''. Do the authors have any clues as to why there is no low-order dependence of the time gap on speed?

In the conclusion, the authors write "In this way, the deduced capacity formula delivers a solid basis to ease the decision-making of operators to find the optimal conveyor speed that is simultaneously compatible with safety, comfort, and performance demands." How do the authors believe that "safety" and "comfort" could be estimated from the simulations?

Reviewer #2: Dear editor,

The manuscript "Come together: A unified description of the escalator capacity" by Christoph Gnendiger and colleagues model passengers dynamics on escalators. The authors model this system using an agent model and obtain an analytical formula for the escalator capacity as a function of the escalator velocity and a T parameter that they interpret as a human reaction time scale. 

In general the paper is interesting, however, I have some reservations about a couple of things. My main issue with the manuscript is that the main finding is the fact that the model agrees with data when using T between 0.15s and 0.30s, which then they conclude to be in agreement with human reaction time from social psychology experiments. However, this is misleading. The model only agrees with the experiments when they artificially expand T as a polynomial function of v_esc. The original model does not agree with the data. They use the 5th power on this expansion without any physical justification. By doing this they are basically fiting C_esc as an inverse power series of v_esc. The authors must make this very clear along the text and abstract.   

Apart from that, I have some small comments:

1) Table 1 is not totally clear for me. The authors refer to "clear width" and here I was lost, because only much later they introduce this parameter. I think they must explain this before, since they will write about this at Table 1. Also, what is "count interval''?

2) In the end of the section "Model of pedestrians", the authors define a set of parameters, a_{agent}=a_{wall}... without any proper explanation. I suppose it is something related with an interaction with the wall, but how? What are the D´s? What are the physical unit of the a´s?

3) On section "Model of escalator" page 5. The authors claim: "Comparing the requirements with Fig. 3, it follows that the deceleration andacceleration of the agent can be adjusted such that it takes place fast enough. "I do not know if this is totally clear. For me this is a strong claim. What led you to claim that? The math being right does not imply that is possible in reality. Perhaps, the authors can clarify this statement. 

4) Reading pag. 7, after Eq. (3), I am not sure if I fully understand the dynamics. Did you define "y" at some previous point? How is the dynamics on y? I think the model in general is not clear regarding the dynamics. 

5) I can not see all the curves in Fig. 5. Are they overlapping? Also, what it is s? Was it properly defined at some point?

6) In general for me it was confusing if I was looking at a continuous model or a discrete one due to the nature of escalators. I think the paper can improve if the authors could separe this better and make clear how the results depend on these two possible conditions.

6. PLOS authors have the option to publish the peer review history of their article (what does this mean?). If published, this will include your full peer review and any attached files.

Reviewer #1: No

Reviewer #2: No

---

## [Author Response · Author response to Decision Letter 0]

13 Dec 2022

The response to specific reviewer comments is provided in the dedicated file 'Response to Reviewers'.

---

## [Decision Letter · Decision Letter 1]

21 Feb 2023

Come together: A unified description of the escalator capacity

PONE-D-22-26795R1

Dear Dr. Gnendiger,

We’re pleased to inform you that your manuscript has been judged scientifically suitable for publication and will be formally accepted for publication once it meets all outstanding technical requirements.

Kind regards,

Ahmed Mancy Mosa, Ph.D.

Academic Editor

PLOS ONE

Additional Editor Comments (optional):

Reviewers' comments:

Reviewer's Responses to Questions

**Comments to the Author**

1. If the authors have adequately addressed your comments raised in a previous round of review and you feel that this manuscript is now acceptable for publication, you may indicate that here to bypass the “Comments to the Author” section, enter your conflict of interest statement in the “Confidential to Editor” section, and submit your "Accept" recommendation.

Reviewer #1: All comments have been addressed

Reviewer #2: All comments have been addressed

2. Is the manuscript technically sound, and do the data support the conclusions?

Reviewer #1: Yes

Reviewer #2: Yes

3. Has the statistical analysis been performed appropriately and rigorously? 

Reviewer #1: Yes

Reviewer #2: N/A

4. Have the authors made all data underlying the findings in their manuscript fully available?

Reviewer #1: (No Response)

Reviewer #2: Yes

5. Is the manuscript presented in an intelligible fashion and written in standard English?

Reviewer #1: Yes

Reviewer #2: Yes

6. Review Comments to the Author

Reviewer #1: All questions and comments were addressed. Therefore, the article is appropriate for publication in PLOS ONE.

Reviewer #2: The authors addressed the main issues, therefore the paper is ready for publication.

7. PLOS authors have the option to publish the peer review history of their article (what does this mean?). If published, this will include your full peer review and any attached files.

Reviewer #1: No

Reviewer #2: No

---

## [Editor Report · Acceptance letter]

24 Feb 2023

PONE-D-22-26795R1 

Come together: A unified description of the escalator capacity 

Dear Dr. Gnendiger:

I'm pleased to inform you that your manuscript has been deemed suitable for publication in PLOS ONE. Congratulations! Your manuscript is now with our production department. 

Kind regards, 

on behalf of

Dr. Ahmed Mancy Mosa 

Academic Editor

PLOS ONE